# Linearly Constrained Weights: Resolving the Vanishing Gradient Problem by Reducing Angle Bias

**Takuro Kutsuna**
Toyota Central R&D Labs, Inc.
Nagakute, Aichi 480-1192, Japan
`kutsuna@mosk.tytlabs.co.jp`

## Abstract

In this paper, we first identify *angle bias*, a simple but remarkable phenomenon that causes the vanishing gradient problem in a multilayer perceptron (MLP) with sigmoid activation functions. We then propose *linearly constrained weights (LCW)* to reduce the angle bias in a neural network, so as to train the network under the constraints that the sum of the elements of each weight vector is zero. A reparameterization technique is presented to efficiently train a model with LCW by embedding the constraints on weight vectors into the structure of the network. Interestingly, batch normalization (Ioffe & Szegedy, 2015) can be viewed as a mechanism to correct angle bias. Preliminary experiments show that LCW helps train a 100-layered MLP more efficiently than does batch normalization.

## 1 Introduction

Neural networks with a single hidden layer have been shown to be universal approximators (Hornik et al., 1989; Irie & Miyake, 1988). However, an exponential number of neurons may be necessary to approximate complex functions. A solution to this problem is to use more hidden layers. The representation power of a network increases exponentially with the addition of layers (Telgarsky, 2016; Eldan & Shamir, 2016). A major obstacle in training deep nets, that is, neural networks with many hidden layers, is the vanishing gradient problem. Various techniques have been proposed for training deep nets, such as layer-wise pretraining (Hinton & Salakhutdinov, 2006), rectified linear units (Nair & Hinton, 2010; Jarrett et al., 2009), variance-preserving initialization (Glorot & Bengio, 2010), and normalization layers (Ioffe & Szegedy, 2015; Gülçehre & Bengio, 2016).

In this paper, we first identify the *angle bias* that arises in the dot product of a nonzero vector and a random vector. The mean of the dot product depends on the angle between the nonzero vector and the mean vector of the random vector. We show that this simple phenomenon is a key cause of the vanishing gradient in a multilayer perceptron (MLP) with sigmoid activation functions. We then propose the use of so-called *linearly constrained weights (LCW)* to reduce the angle bias in a neural network. LCW is a weight vector subject to the constraint that the sum of its elements is zero. A reparameterization technique is presented to embed the constraints on weight vectors into the structure of a neural network. This enables us to train a neural network with LCW by using optimization solvers for unconstrained problems, such as stochastic gradient descent. Preliminary experiments show that we can train a 100-layered MLP with sigmoid activation functions by reducing the angle bias in the network. Interestingly, batch normalization (Ioffe & Szegedy, 2015) can be viewed as a mechanism to correct angle bias in a neural network, although it was originally developed to overcome another problem, that is, the internal covariate shift problem. Preliminary experiments suggest that LCW helps train deep MLPs more efficiently than does batch normalization.

In Section 2, we define angle bias and discuss its relation to the vanishing gradient problem. In Section 3, we propose LCW as an approach to reduce angle bias in a neural network. We also present a reparameterization technique to efficiently train a model with LCW and an initialization method for LCW. In Section 4, we review related work; mainly, we examine existing normalization techniques from the viewpoint of reducing the angle bias. In Section 5, we present empirical results

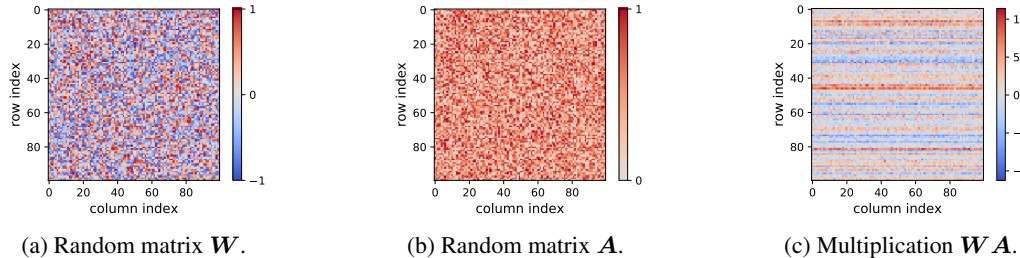

(a) Random matrix $\boldsymbol{W}$.          (b) Random matrix $\boldsymbol{A}$.          (c) Multiplication $\boldsymbol{WA}$.

Figure 1: Angle bias causes a horizontal stripe pattern in $\boldsymbol{WA}$. (Best viewed in color)

that show that it is possible to efficiently train a 100-layered MLP by reducing the angle bias using LCW. Finally, we conclude with a discussion of future works.

## 2 ANGLE BIAS

We introduce angle bias by using the simple example shown in Figure 1. Figure 1(a) is a heat map representation of matrix $\boldsymbol{W} \in R^{100 \times 100}$, each of whose elements is independently drawn from a uniform random distribution in the range $(-1, 1)$. Matrix $\boldsymbol{A} \in R^{100 \times 100}$ is also generated randomly, and its elements range from 0 to 1, as shown in Figure 1(b). We multiply $\boldsymbol{W}$ and $\boldsymbol{A}$ to obtain the matrix shown in Figure 1(c). Unexpectedly, a horizontal stripe pattern appears in the heat map of $\boldsymbol{WA}$ although both $\boldsymbol{W}$ and $\boldsymbol{A}$ are random matrices. This pattern is attributed to the angle bias that is defined as follows:

**Definition 1.** $\mathcal{P}_\gamma$ is an $m$ dimensional probability distribution whose expected value is $\gamma \mathbf{1}_m$, where $\gamma \in R$ and $\mathbf{1}_m$ is an $m$ dimensional vector whose elements are all one.

**Proposition 1.** Let $\boldsymbol{a}$ be a random vector in $R^m$ that follows $\mathcal{P}_\gamma$. Given $\boldsymbol{w} \in R^m$ such that $\|\boldsymbol{w}\| > 0$, the expected value of $\boldsymbol{w} \cdot \boldsymbol{a}$ is $|\gamma|\sqrt{m}\|\boldsymbol{w}\| \cos\theta_w$, where $\theta_w$ is the angle between $\boldsymbol{w}$ and $\mathbf{1}_m$.

*Proof.* It follows from $E(\boldsymbol{w} \cdot \boldsymbol{a}) = \boldsymbol{w} \cdot E(\boldsymbol{a}) = \boldsymbol{w} \cdot (\gamma \mathbf{1}_m) = \|\boldsymbol{w}\|\ \|\gamma \mathbf{1}_m\| \cos\theta_w = |\gamma|\sqrt{m}\|\boldsymbol{w}\| \cos\theta_w$, where $E(x)$ denotes the expected value of random variable $x$. $\qquad\square$

**Definition 2.** From Proposition 1, the expected value of $\boldsymbol{w} \cdot \boldsymbol{a}$ depends on $\theta_w$ as long as $\gamma \neq 0$. The distribution of $\boldsymbol{w} \cdot \boldsymbol{a}$ is then biased depending on $\theta_w$; this is called *angle bias*.

In Figure 1, if we denote the $i$-th row vector of $\boldsymbol{W}$ and the $j$-th column vector of $\boldsymbol{A}$ by $\boldsymbol{w}_i$ and $\boldsymbol{a}_j$, respectively, $\boldsymbol{a}_j$ follows $\mathcal{P}_\gamma$ with $\gamma = 0.5$. The $i$-th row of $\boldsymbol{WA}$ is biased according to the angle between $\boldsymbol{w}_i$ and $\mathbf{1}_m$, because the $(i, j)$-th element of $\boldsymbol{WA}$ is the dot product of $\boldsymbol{w}_i$ and $\boldsymbol{a}_j$. Note that if the random matrix $\boldsymbol{A}$ has both positive and negative elements, $\boldsymbol{WA}$ also shows a stripe pattern as long as each column vector of $\boldsymbol{A}$ follows $\mathcal{P}_\gamma$ with $\gamma \neq 0$.

We can generalize Proposition 1 for any $m$ dimensional distribution $\hat{\mathcal{P}}$, instead of $\mathcal{P}_\gamma$, as follows:

**Proposition 2.** Let $\hat{\boldsymbol{a}}$ be a random vector that follows an $m$ dimensional probability distribution $\hat{\mathcal{P}}$ whose expected value is $\hat{\boldsymbol{\mu}} \in R^m$. Given $\boldsymbol{w} \in R^m$ such that $\|\boldsymbol{w}\| > 0$, it follows that

$$E(\boldsymbol{w} \cdot \hat{\boldsymbol{a}}) = \begin{cases} \|\boldsymbol{w}\|\ \|\hat{\boldsymbol{\mu}}\| \cos\hat{\theta}_w & \text{if } \|\hat{\boldsymbol{\mu}}\| > 0, \\ 0 & \text{otherwise,} \end{cases}$$

where $\hat{\theta}_w$ is the angle between $\boldsymbol{w}$ and $\hat{\boldsymbol{\mu}}$.

*Proof.* The proof is the same as that of Proposition 1. $\qquad\square$

Proposition 2 states that the distribution of $\boldsymbol{w} \cdot \hat{\boldsymbol{a}}$ is biased according to $\hat{\theta}_w$ unless $\|\hat{\boldsymbol{\mu}}\| = 0$.

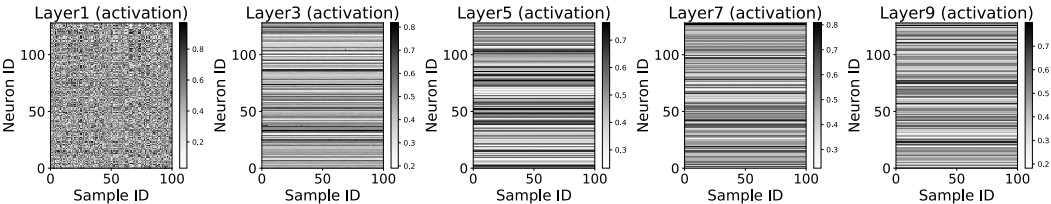

Figure 2: Activations in layers 1, 3, 5, 7, and 9 of the 10 layered MLP with sigmoid activation functions. Inputs are randomly sampled from CIFAR-10.

## 2.1 ANGLE BIAS IN A MULTILAYER PERCEPTRON

We consider a standard MLP. For simplicity, the number of neurons $m$ is assumed to be the same in all layers. The activation vector in layer $l$ is denoted by $\boldsymbol{a}^l = \left(a_1^l, \ldots, a_m^l\right)^\top \in R^m$. The weight vector of the $i$-th neuron in layer $l$ is denoted by $\boldsymbol{w}_i^l \in R^m$. It is generally assumed that $\|\boldsymbol{w}_i^l\| > 0$. The activation of the $i$-th neuron in layer $l$ is given by

$$a_i^l = f\left(z_i^l\right), \tag{1}$$
$$z_i^l = \boldsymbol{w}_i^l \cdot \boldsymbol{a}^{l-1} + b_i^l \tag{2}$$
$$= \|\boldsymbol{w}_i^l\| \, \|\boldsymbol{a}^{l-1}\| \cos\theta_i^l + b_i^l, \tag{3}$$

where $f$ is a nonlinear activation function, $b_i^l \in R$ is the bias term, $z_i^l \in R$ denotes the preactivation value, and $\theta_i^l \in R$ means the angle between $\boldsymbol{w}_i^l$ and $\boldsymbol{a}^{l-1}$. We assume that $\boldsymbol{a}^0$ corresponds to the input vector to the MLP. In Equations (1), (2), and (3), variables $z_i^l$, $a_i^l$, and $\theta_i^l$ are regarded as random variables whose distributions are determined by the distribution of the input vector $\boldsymbol{a}^0$, given the weight vectors and the bias terms.

From Proposition 2 and Equation (2), if $\boldsymbol{a}^{l-1}$ follows $\hat{\mathcal{P}}$ with the mean vector $\hat{\boldsymbol{\mu}}^{l-1}$ such that $\|\hat{\boldsymbol{\mu}}^{l-1}\| > 0$, the preactivation $z_i^l$ is biased according to the angle between $\boldsymbol{w}_i^l$ and $\hat{\boldsymbol{\mu}}^{l-1}$. If $f$ is assumed to be the sigmoid activation function, $\boldsymbol{a}^l$ is then biased toward a specific region in $(0,1)^m$, because each element of $\boldsymbol{a}^l$ have a different mean value from Equation (1). In the next layer, the variance of $\theta_i^{l+1}$ will shrink compared to that of $\theta_i^l$, because $\boldsymbol{a}^l$ is biased in $(0,1)^m$. The variance of $a_i^{l+1}$ shrinks as the variance of $\theta_i^{l+1}$ shrinks from Equations (1) and (3). Repeating the operations through multiple layers, the variance of $\theta_i^l$ and $a_i^l$ will shrink to small values.

## 2.2 VISUALIZING EFFECT OF ANGLE BIAS USING CIFAR-10 DATASET

We illustrate the effect of angle bias in an MLP by using the CIFAR-10 dataset (Krizhevsky & Hinton, 2009) that includes a set of $32 \times 32$ color (RGB) images. Each sample in CIFAR-10 is considered an input vector with $32 \times 32 \times 3 = 3072$ real values, in which each variable is scaled into the range $[-1, 1]$.

### 2.2.1 WITH SIGMOID ACTIVATION FUNCTIONS

We consider an MLP with sigmoid activation functions that has 10 hidden layers with $m = 128$ neurons in each layer. The weights of the MLP are initialized according to Glorot & Bengio (2010). We randomly took 100 samples from the dataset and input them into the MLP. Figure 2 shows the activation pattern in layers 1, 3, 5, 7, and 9 on the selected samples. Please note that the activation in Layer 1 corresponds to $a_i^1$ in Equation (1), that is, Layer 1 is the layer after the input layer. We see stripe patterns in the layers other than Layers 1 in Figure 2 that are caused by angle bias. In Layer 9, the activation value of each neuron is almost constant regardless of the input. In contrast, no stripe pattern appears in Layer 1, because each element of the input vector is scaled into the range $[-1, 1]$ and its mean value is near zero; this corresponds to the case in which $\|\hat{\boldsymbol{\mu}}\| \approx 0$ in Proposition 2.

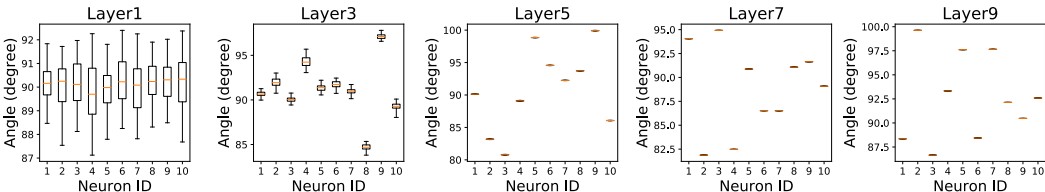

Figure 3: Boxplot summaries of $\theta_i^l$ in the MLP with sigmoid activation functions. Results for only the first ten neurons in layers 1,3,5,7, and 9 are displayed. Samples shown in Figure 2 are used to evaluate $\theta_i^l$.

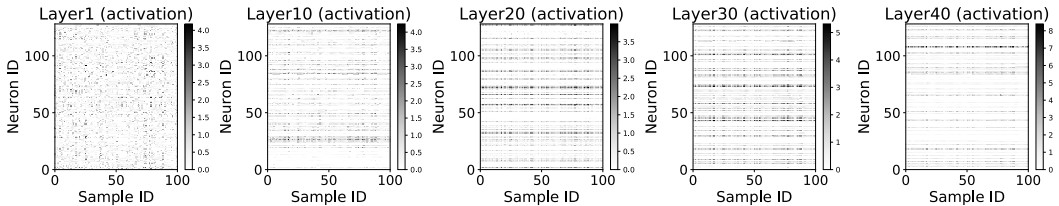

Figure 4: Activations in layers 1, 10, 20, 30, and 40 of the 50 layered MLP with ReLU activation functions. Inputs are randomly sampled from CIFAR-10.

We calculated the angle between $\boldsymbol{w}_i^l$ and $\boldsymbol{a}^{l-1}$, that is, $\theta_i^l$ in Equation (3), for each sample[1]. Figure 3 shows boxplot summaries of $\theta_i^l$ on the first ten neurons in layers 1, 3, 5, 7, and 9, in which the 1%, 25%, 50%, 75%, and 99% quantiles are displayed as whiskers or boxes. We see the mean of $\theta_i^l$ are biased according to the neurons in the layers other than Layer 1. We also see that the variance of $\theta_i^l$ shrink through layers.

### 2.2.2 WITH ReLU ACTIVATION FUNCTIONS

Next, we consider an MLP with ReLU activation functions that has 50 hidden layers with $m = 128$ neurons in each layer. The weights are initialized according to Glorot & Bengio (2010). Figure 4 shows the activation pattern in layers 1, 10, 20, 30, and 40 on the randomly selected samples. We see stripe patterns in the layers other than Layer 1 that are caused by the angle bias.

Figure 5 shows boxplot summaries of $\theta_i^l$ on the first ten neurons in layers 1, 10, 20, 30, and 40. We see that the mean of $\theta_i^l$ are biased according the neurons in the layers other than Layer 1. We also see that the variance of $\theta_i^l$ shrink through layers, but the shrinking rate is much moderate compared to that in Figure 3. This is because ReLU projects a preactivation vector into the unbounded region $[0, +\infty)^m$ and the activation vector is less likely to concentrate on a specific region.

### 2.3 RELATION TO VANISHING GRADIENT PROBLEM

Under the effect of angle bias, the activation of neurons in deeper layers are almost constant regardless of the input in an MLP with sigmoid activation functions, as shown in Figure 2. It indicates that $\nabla_{\boldsymbol{a}^0} L = \boldsymbol{0}$, where $L$ is a loss function that is defined based on the output of the MLP and $\nabla_{\boldsymbol{a}^0} L$ means the gradient with respect to the input vector $\boldsymbol{a}^0$. From Equation (2), we have

$$\nabla_{\boldsymbol{a}^{l-1}} L = \sum_{i=1}^m \boldsymbol{w}_i^l \nabla_{z_i^l} L, \tag{4}$$

$$\nabla_{\boldsymbol{w}_i^l} L = \boldsymbol{a}^{l-1} \nabla_{z_i^l} L \quad (i = 1, \dots, m). \tag{5}$$

Assuming that $\boldsymbol{w}_i^1 \, (i = 1, \dots, m)$ are linearly independent, it follows that $\nabla_{z_i^1} = 0 \, (i = 1, \dots, m)$ from Equation (4), with $l = 1$, and $\nabla_{\boldsymbol{a}^0} L = \boldsymbol{0}$. Then, it holds that $\nabla_{\boldsymbol{w}_i^1} L = 0 \, (i = 1, \dots, m)$ from

---

[1] The angle is given by $\arccos\left(\left(\boldsymbol{w}_i^l \cdot \boldsymbol{a}^{l-1}\right) / \left(\|\boldsymbol{w}_i^l\| \, \|\boldsymbol{a}^{l-1}\|\right)\right)$

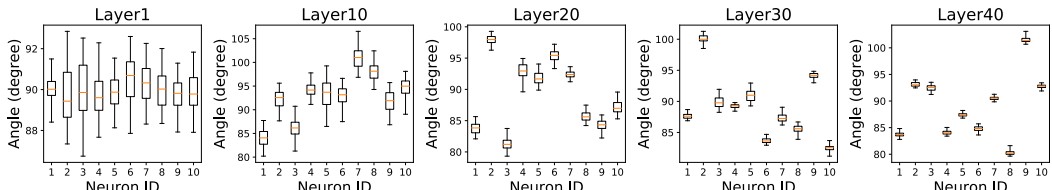

Figure 5: Boxplot summaries of $\theta_i^l$ in the MLP with ReLU activation functions. Results for only the first ten neurons in layers 1, 10, 20, 30, and 40 are displayed. Samples shown in Figure 4 are used to evaluate $\theta_i^l$.

Equation (5), with $l = 1$, indicating that the gradients of weights in the first layer are vanished. From Equation (1), with $l = 1$, we have $\nabla_{z_i^1} L = f'(z_i^1) \nabla_{a_i^1} L$. If $f'(z_i^1) \neq 0$, it follows that $\nabla_{a_i^1} L = 0$ from $\nabla_{z_i^1} L = 0$. This leads to $\nabla_{\boldsymbol{w}_i^2} L = 0$ from Equations (4) and (5), with $l = 2$, under the assumption that $\boldsymbol{w}_i^2$ $(i = 1, \ldots, m)$ are linearly independent. Consequently, we can derive $\nabla_{\boldsymbol{w}_i^l} L = 0$ from $\nabla_{\boldsymbol{a}^0} L = \boldsymbol{0}$ so far as $f'(z_i^l) \neq 0$ and $\boldsymbol{w}_i^l$ $(i = 1, \ldots, m)$ being linearly independent.

If we use rectified linear activation instead of sigmoid activation, the gradients of weights are less likely to vanish, because $\nabla_{\boldsymbol{a}^0} L$ will seldom be exactly zero. However, the rate of each neuron being active[2] is biased, because the distribution of preactivation $z_i^l$ is biased. If a neuron is always active, it behaves as an identity mapping. If a neuron is always inactive, it is worthless because its output is always zero. Such a phenomenon is observed in deep layers in Figure 4. As discussed in Balduzzi et al. (2017), the efficiency of the network decreases in this case. In this sense, angle bias may reduce the efficiency of a network with rectified linear activation.

## 3 LINEARLY CONSTRAINED WEIGHTS

There are two approaches to reduce angle bias in a neural network. The first one is to somehow make the expected value of the activation of each neuron near zero, because angle bias does not occur if $\|\hat{\boldsymbol{\mu}}\| = 0$ from Proposition 2. The second one is to somehow regularize the angle between $\boldsymbol{w}_i^l$ and $E\left(\boldsymbol{a}^{l-1}\right)$. In this section, we propose a method to reduce angle bias in a neural network by using the latter approach. We introduce $\mathcal{W}_{\text{LC}}$ as follows:

**Definition 3.** $\mathcal{W}_{\text{LC}}$ is a subspace in $R^m$ defined by

$$\mathcal{W}_{\text{LC}} := \left\{ \boldsymbol{w} = (w_1, \ldots, w_m)^\top \in R^m \ \middle| \ w_1 + \ldots + w_m = 0 \right\}. \tag{6}$$

The following holds for $\boldsymbol{w} \in \mathcal{W}_{\text{LC}}$:

**Proposition 3.** Let $\boldsymbol{a}$ be an $m$ dimensional random variable that follows $\mathcal{P}_\gamma$. Given $\boldsymbol{w} \in \mathcal{W}_{\text{LC}}$ such that $\|\boldsymbol{w}\| > 0$, the expected value of $\boldsymbol{w} \cdot \boldsymbol{a}$ is zero.

*Proof.* $E(\boldsymbol{w} \cdot \boldsymbol{a}) = \boldsymbol{w} \cdot E(\boldsymbol{a}) = \gamma \left(\boldsymbol{w} \cdot \boldsymbol{1}_m\right) = \gamma \left(w_1 + \ldots + w_m\right) = 0.$ □

From Proposition 3, if $\boldsymbol{w}_i^l \in \mathcal{W}_{\text{LC}}$ and $\boldsymbol{a}^{l-1}$ follows $\mathcal{P}_\gamma$, we can resolve angle bias in $z_i^l$ in Equation (2). If we initialize $b_i^l = 0$, the distribution of each of $z_i^l$ $(i = 1, \ldots, m)$ will likely be more similar to each other. The activation vector in layer $l$, each of whose elements is given by Equation (1), is then expected to follow $\mathcal{P}_\gamma$. Therefore, if the input vector $\boldsymbol{a}^0$ follows $\mathcal{P}_\gamma$, we can inductively reduce the angle bias in each layer of an MLP by using weight vectors that are included in $\mathcal{W}_{\text{LC}}$. We call weight vector $\boldsymbol{w}_i^l$ in $\mathcal{W}_{\text{LC}}$ *linearly constrained weights (LCWs)*.

---

[2] A neuron with rectified linear activation is said to be *active* if its output value is positive.

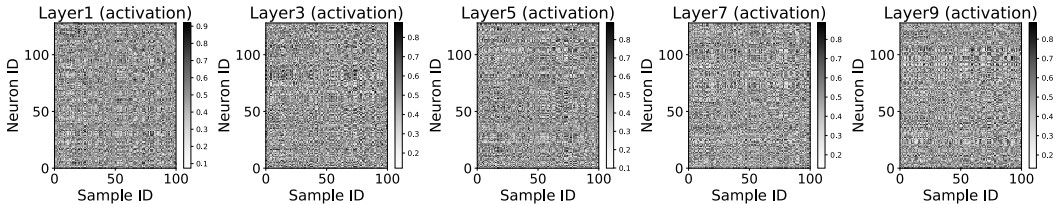

Figure 6: Activations in layers 1, 3, 5, 7, and 9 of the MLP with sigmoid activation functions with LCW. The input samples are the same as those used in Figure 2.

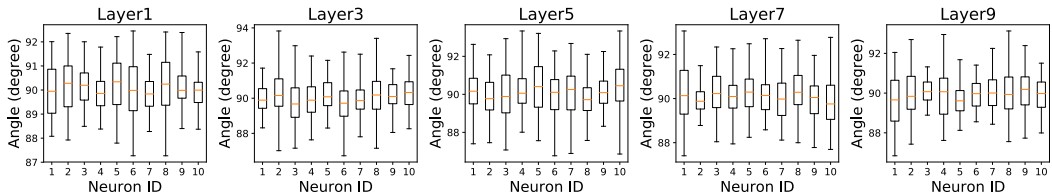

Figure 7: Boxplot summaries of $\theta_i^l$ on the first ten neurons in layers 1,3,5,7, and 9 of the MLP with sigmoid activation functions with LCW.

## 3.1 VISUALIZING THE EFFECT OF LCW

### 3.1.1 WITH SIGMOID ACTIVATION FUNCTIONS

We built an MLP with sigmoid activation functions of the same size as that used in Section 2.2.1, but whose weight vectors are replaced with LCWs. We applied the minibatch-based initialization described in Section 3.3. Figure 6 shows the activation pattern in layers 1, 3, 5, 7, and 9 of the MLP with LCW on the randomly selected samples that are used in Figure 2. When compared with Figure 2, we see no stripe pattern in Figure 6. The neurons in Layer 9 respond differently to each input sample; this means that a change in the input leads to a different output. Therefore, the network output changes if we adjust the weight vectors in Layer 1, that is, the gradients of weights in Layer 1 do not vanish in Figure 6.

Figure 7 shows boxplot summaries of $\theta_i^l$ on the first ten neurons in layers 1, 3, 5, 7, and 9 of the MLP with LCW. We see that the angle distributes around 90° on each neuron in each layer. This indicates that the angle bias is resolved in the calculation of $z_i^l$ by using LCW.

**After 10 epochs training** Figure 8 shows the activation pattern in layers of the MLP with LCW after 10 epochs training. A slight stripe pattern is visible in Figure 8, but neurons in each layer react differently to each input. Figure 9 shows boxplot summaries of $\theta_i^l$ of the MLP after 10 epochs training. We see that the mean of $\theta_i^l$ is slightly biased according to the neurons. However, the variance of $\theta_i^l$ do not shrink even in deeper layers.

### 3.1.2 WITH RELU ACTIVATION FUNCTIONS

We built an MLP with ReLU activation functions of the same size as that used in Section 2.2.2, whose weight vectors are replaced with LCWs. We applied the minibatch-based initialization described in Section 3.3. Figure 10 shows the activation pattern in layers 1, 10, 20, 30, and 40 of the MLP with LCW. When compared with Figure 4, we see no stripe pattern in Figure 10. Figure 11 shows boxplot summaries of $\theta_i^l$ on the first ten neurons in layers 1, 10, 20, 30, and 40 of the MLP with LCW. We can observe that the angle bias is resolved by using LCW in the MLP with ReLU activation functions.



Figure 8: Activations in layers of the MLP with sigmoid activation functions with LCW after 10 epochs training.

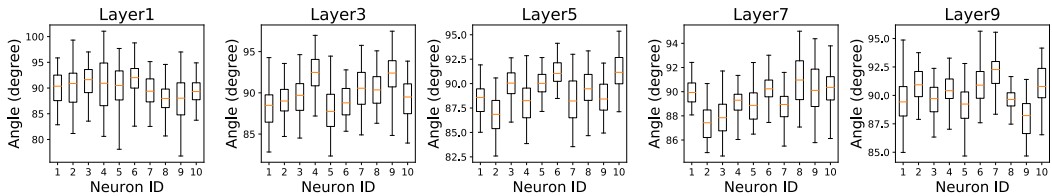

Figure 9: Boxplot summaries of $\theta_i^l$ of the MLP with sigmoid activation functions with LCW after 10 epochs training.

## 3.2 LEARNING LCW VIA REPARAMETERIZATION

A straightforward way to train a neural network with LCW is to solve a constrained optimization problem, in which a loss function is minimized under the condition that each weight vector is included in $\mathcal{W}_{\mathrm{LC}}$. Although several methods are available to solve such constrained problems, for example, the gradient projection method (Luenberger & Ye, 2015), it might be less efficient to solve a constrained optimization problem than to solve an unconstrained one. We propose a reparameterization technique that enables us to train a neural network with LCW by using a solver for unconstrained optimization. We can embed the constraints on the weight vectors into the structure of the neural network by reparameterization.

**Reparameterization** Let $\boldsymbol{w}_i^l \in R^m$ be a weight vector in a neural network. We reparametrize $\boldsymbol{w}_i^l$ by using vector $\boldsymbol{v}_i^l \in R^{m-1}$ as $\boldsymbol{w}_i^l = \boldsymbol{B}_m \boldsymbol{v}_i^l$, where $\boldsymbol{B}_m \in R^{m \times (m-1)}$ is a basis of $\mathcal{W}_{\mathrm{LC}}$, written as a matrix of column vectors. For example, $\boldsymbol{B}_m$ is given by

$$\boldsymbol{B}_m = \begin{pmatrix} \boldsymbol{I}_{m-1} \\ -\boldsymbol{1}_{m-1}^\top \end{pmatrix} \in R^{m \times (m-1)}, \tag{7}$$

where $\boldsymbol{I}_{m-1}$ is the identity matrix of order $(m-1) \times (m-1)$.

It is obvious that $\boldsymbol{w}_i^l = \boldsymbol{B}_m \boldsymbol{v}_i^l \in \mathcal{W}_{\mathrm{LC}}$. We then solve the optimization problem in which $\boldsymbol{v}_i^l$ is considered as a new variable in place of $\boldsymbol{w}_i^l$. This optimization problem is unconstrained because $\boldsymbol{v}_i^l \in R^{m-1}$. We can search for $\boldsymbol{w}_i^l \in \mathcal{W}_{\mathrm{LC}}$ by exploring $\boldsymbol{v}_i^l \in R^{m-1}$. In the experiments in Section 5, we used $\boldsymbol{B}_m$ in Equation (7). We also tried an orthonormal basis of $\mathcal{W}_{\mathrm{LC}}$ as $\boldsymbol{B}_m$; however, there was little difference in accuracy. It is worth noting that the proposed reparameterization can be implemented easily and efficiently by using modern frameworks for deep learning using GPUs.

## 3.3 INITIALIZATION USING MINIBATCH STATISTICS

By introducing LCW, we can reduce the angle bias in $z_i^l$ in Equation (2), which mainly affects the expected value of $z_i^l$. It is also important to regularize the variance of $z_i^l$, especially when the sigmoid activation is used, because the output of the activation will likely saturate when the variance of $z_i^l$ is too large. We apply an initialization method by which the variance of $z_i^l$ is regularized based on a minibatch of samples. This type of initialization has also been used in previous studies Mishkin & Matas (2016) and Salimans & Kingma (2016).

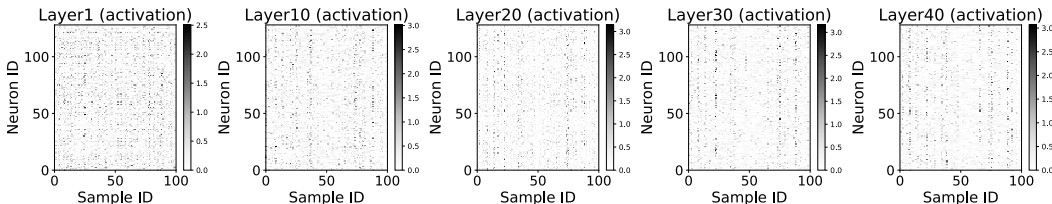

Figure 10: Activations in layers 1, 10, 20, 30, and 40 of the MLP with ReLU activation functions with LCW. The input samples are the same as those used in Figure 4.

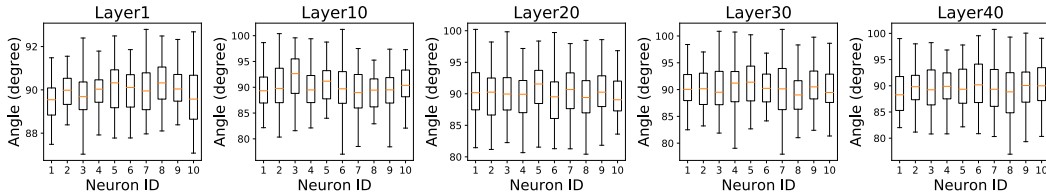

Figure 11: Boxplot summaries of $\theta_i^l$ on the first ten neurons in layers 1, 10, 20, 30, and 40 of the MLP with ReLU activation functions with LCW.

First, we randomly generate a seed vector $\hat{\boldsymbol{w}}_i^l \in \mathcal{W}_{\mathrm{LC1}}$, where $\mathcal{W}_{\mathrm{LC1}} := \left\{ \boldsymbol{w} \in \mathcal{W}_{\mathrm{LC}} \mid \|\boldsymbol{w}\| \leq 1 \right\}$. Section A in the appendix describes the approach for sampling uniformly from $\mathcal{W}_{\mathrm{LC1}}$. The scholar variable $\eta^l$ is then calculated in a layer-wise manner with the property that the standard deviation of $\left(\eta^l \hat{\boldsymbol{w}}_i^l\right) \cdot a^{l-1}$ equals the target value $\sigma_z$, in which $a^{l-1}$ is evaluated using a minibatch. Finally, we use $\eta^l \hat{\boldsymbol{w}}_i^l$ as the initial value of $\boldsymbol{w}_i^l$. If we apply the reparameterization method described in the previous section, the first $m-1$ elements of $\eta^l \hat{\boldsymbol{w}}_i^l$ are used as the initial value of $\boldsymbol{v}_i^l$. We initialize bias terms $b_i^l$ in Equation (2) to zero.

## 4 RELATED WORK

Ioffe & Szegedy (2015) proposed the *batch normalization (BN)* approach for accelerating the training of deep nets. BN was developed to address the problem of *internal covariate shift*, that is, training deep nets is difficult because the distribution of the input of a layer changes as the weights of the preceding layers change during training. The computation of the mean and standard deviation of $z_i^l$ based on a minibatch is incorporated into the structure of the network, and $z_i^l$ is normalized by using these statistics. Gülçehre & Bengio (2016) proposed the *standardization layer (SL)* approach, which is similar to BN. The main difference is that SL normalizes $a_i^l$, whereas BN normalizes $z_i^l$. Interestingly, both BN and SL can be considered mechanisms for reducing the angle bias. SL reduces the angle bias by forcing $\|\hat{\boldsymbol{\mu}}\| = 0$ in Proposition 2. On the other hand, BN reduces the angle bias by normalizing $z_i^l$ for each neuron. A drawback of both BN and SL is that the model has to be switched during inference to ensure that its output depends only on the input but not the minibatch. By contrast, LCW proposed in this paper does not require any change in the model during inference. Moreover, the computational overhead for training the LCW is lower than that of BN and SL, because LCW only requires the addition of layers that multiply $\boldsymbol{B}_m$ to $\boldsymbol{v}_i^l$, as described in Section 3.2, whereas BN and SL need to compute both the mean and the standard deviation of $z_i^l$ or $a_i^l$.

Salimans & Kingma (2016) proposed *weight normalization (WN)* to overcome the drawbacks of BN, in which a weight vector $\boldsymbol{w}_i^l \in R^m$ is reparametrized as $\boldsymbol{w}_i^l = (g_i^l / \|\boldsymbol{v}_i^l\|) \boldsymbol{v}_i^l$, where $g_i^l \in R$ and $\boldsymbol{v}_i^l \in R^m$ are new parameters. By definition, WN does not have the property of reducing the angle bias, because the degrees of freedom of $\boldsymbol{w}_i^l$ are unchanged by the reparameterization. Salimans & Kingma (2016) also proposed a minibatch-based initialization by which weight vectors are initialized so that $z_i^l$ has zero mean and unit variance, indicating that the angle bias is reduced immediately after the initialization.

Ba et al. (2016) proposed *layer normalization (LN)* as a variant of BN. LN normalizes $z_i^l$ over the neurons in a layer on a sample in the minibatch, whereas BN normalizes $z_i^l$ over the minibatch on a neuron. From the viewpoint of reducing the angle bias, LN is not as direct as BN. Although LN does not resolve the angle bias, it is expected to normalize the degree of bias in each layer.

## 5 EXPERIMENTS

We conducted preliminary experiments using the CIFAR-10 dataset, the CIFAR-100 dataset (Krizhevsky & Hinton, 2009), and the SVHN dataset (Netzer et al., 2011). These experiments are aimed not at achieving state-of-the-art results but at investigating whether we can train a deep model by reducing the angle bias and empirically evaluating the performance of LCW in comparison to that of BN and WN.

**Network structure**    We used MLPs with the cross-entropy loss function. Each network has $32 \times 32 \times 3 = 3072$ input neurons and 10 output neurons, and it is followed by a softmax layer. We refer to an MLP that has $L$ hidden layers and $M$ neurons in each hidden layer as $\mathrm{MLP}(L, M)$. Either a sigmoid activation function or a rectified linear activation function was used. $\mathrm{MLP_{LCW}}$ denotes an MLP in which each weight vector is replaced by LCW. $\mathrm{MLP_{BN}}$ denotes an MLP in which the preactivation of each neuron is normalized by BN. $\mathrm{MLP_{WN}}$ denotes an MLP whose weight vectors are reparametrized by WN.

**Initialization**    Plain MLP and $\mathrm{MLP_{BN}}$ were initialized using the method proposed in Glorot & Bengio (2010). $\mathrm{MLP_{LCW}}$ was initialized using the minibatch-based method described in Section 3.3 with $\sigma_z = 0.5$. $\mathrm{MLP_{WN}}$ was initialized according to Salimans & Kingma (2016).

**Optimization**    MLPs were trained using a stochastic gradient descent with a minibatch size of 128 for 100 epochs. The learning rate starts from 0.1 and is multiplied by 0.95 after every two epochs.

**Environment and implementation**    The experiments were performed on a system running Ubuntu 16.04 LTS with NVIDIA® Tesla® K80 GPUs. We implemented LCW using PyTorch version 0.1.12. We implemented BN using the `torch.nn.BatchNorm1d` module in PyTorch. We implemented WN by ourselves using PyTorch[3].

### 5.1 EXPERIMENTAL RESULTS: MLPS WITH SIGMOID ACTIVATION FUNCTIONS

We first consider MLPs with sigmoid activation functions. Figure 12 shows the convergence and computation time for training MLPs with CIFAR-10 dataset. Figure 12(a)-(c) shows results for $\mathrm{MLP}(100, 128)$, Figure 12(d)-(f) for $\mathrm{MLP}(50, 256)$, and Figure 12(g)-(i) for $\mathrm{MLP}(5, 512)$. Figure 12(a) shows that the training accuracy of the plain $\mathrm{MLP}(100, 128)$ is 10% throughout the training, because the MLP output is insensible to the input because of the angle bias, as mentioned in Section 2.2[4]. By contrast, $\mathrm{MLP_{LCW}}$ or $\mathrm{MLP_{BN}}$ is successfully trained, as shown in Figure 12(a), indicating that the angle bias is a crucial obstacle to training deep MLPs with sigmoid activation functions. $\mathrm{MLP_{LCW}}$ achieves a higher rate of increase in the training accuracy compared to $\mathrm{MLP_{BN}}$ in Figure 12(a), (d), and (g). As described in Section 4, WN itself cannot reduce the angle bias, but the bias is reduced immediately after the initialization of WN. From Figure 12(a) and (d), we see that deep MLPs with WN are not trainable. These results suggest that starting with weight vectors that do not incur angle bias is not sufficient to train deep nets. It is important to incorporate a mechanism that reduces the angle bias during training, such as LCW or BN.

The computational overhead of training of $\mathrm{MLP_{LCW}}(100, 128)$ is approximately 55% compared to plain $\mathrm{MLP}(100, 128)$, as shown in Figure 12(b); this is much lower than that of $\mathrm{MLP_{BN}}(100, 128)$. The overhead of $\mathrm{MLP_{WN}}$ is large compared to that of $\mathrm{MLP_{BN}}$, although it contradicts the claim of Salimans & Kingma (2016). We think this is due to the implementation of these methods. The BN

---

[3]Although a module for WN is available in PyTorch version 0.2.0, we did not use it because the minibatch-based initialization (Salimans & Kingma, 2016) is not implemented for this module.

[4]CIFAR-10 includes samples from 10 classes equally. The prediction accuracy is therefore 10% if we predict all samples as the same class.

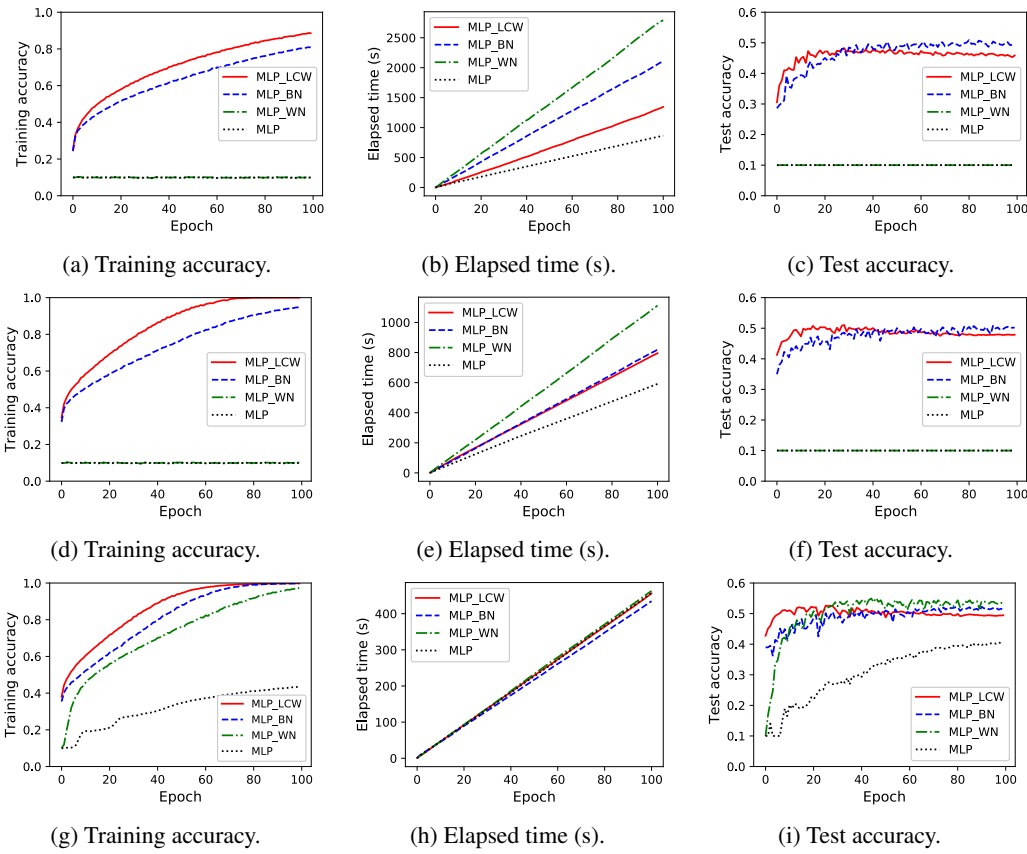

Figure 12: Training accuracy, elapsed time, and test accuracy for CIFAR-10 dataset: (a-c) results of $\text{MLP}(100, 128)$, (d-f) results of $\text{MLP}(50, 256)$, (g-i) results of $\text{MLP}(5, 512)$. Sigmoid activation functions are used in each model.

module we used in the experiments consists of a specialized function developed by GPU vendors, whereas the WN module was developed by ourselves. In this sense, the overhead of LCW may be improved by a more sophisticated implementation.

In terms of the test accuracy, $\text{MLP}_{\text{LCW}}$ has peaks around 20 epochs, as shown in Figure 12(c), (f), and (i). We have no clear explanation for this finding, and further studies are needed to investigate the generalizability of neural networks.

Experimental results with SVHN and CIFAR-100 datasets are reported in Section B in the appendix.

## 5.2 Experimental results: MLPs with rectified linear activation functions

We have experimented with MLPs with rectified linear activation functions. In our experiments, we observed that the plain MLP with 20 layers and 256 neurons per layer was successfully trained. However, the training of $\text{MLP}_{\text{LCW}}$ of the same size did not proceed at all, regardless of the dataset used in our experiment; in fact, the output values of the network exploded after a few minibatch updates. We have investigated the weight gradients of the plain MLP and $\text{MLP}_{\text{LCW}}$. Figure 13 shows boxplot summaries of the weight gradients in each layer of both models, in which the gradients are evaluated by using a minibatch of CIFAR-10 immediately after the initialization. By comparing Figure 13(a) and Figure 13(b), we find an exponential increase in the distributions of the weight gradients of $\text{MLP}_{\text{LCW}}$ in contrast to the plain MLP. Because the learning rate was the same for every layer in our experiments, this exponential increase of the gradients might hamper the learning of $\text{MLP}_{\text{LCW}}$. The gradients in a rectifier network are sums of path-weights over *active paths* (Balduzzi et al., 2017). The exponential increase of the gradients therefore implies an exponential increase

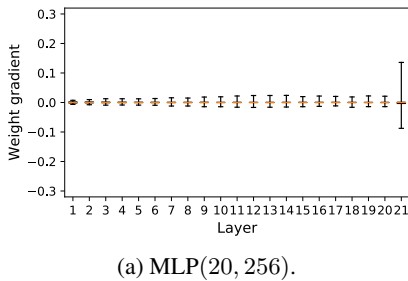

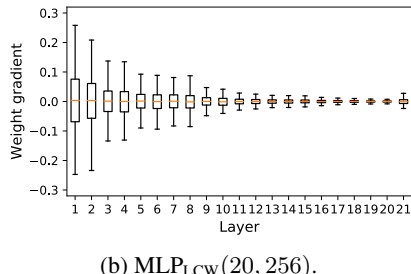

(a) MLP(20, 256).          (b) MLP$_{LCW}$(20, 256).

Figure 13: Distributions of weight gradients in each layer of MLP(20, 256) or MLP$_{LCW}$(20, 256) with rectified linear activation functions.

of active paths. As discussed in Section 2.3, we can prevent neurons from being *always inactive* by reducing the angle bias, which we think caused the exponential increase in active paths. We need further studies to make MLP$_{LCW}$ with rectified linear activation functions trainable. Possible directions are to apply layer-wise learning rates or to somehow regularize the distribution of the weight gradients in each layer of MLP$_{LCW}$, which we leave as future work.

## 6 CONCLUSIONS AND FUTURE WORK

In this paper, we have first identified the angle bias that arises in the dot product of a nonzero vector and a random vector. The mean of the dot product depends on the angle between the nonzero vector and the mean vector of the random vector. In a neural network, the preactivation value of a neuron is biased depending on the angle between the weight vector of the neuron and the mean of the activation vector in the previous layer. We have shown that such biases cause a vanishing gradient in a neural network with sigmoid activation functions. To overcome this problem, we have proposed linearly constrained weights to reduce the angle bias in a neural network; these can be learned efficiently by the reparameterization technique. Preliminary experiments suggest that reducing the angle bias is essential to train deep MLPs with sigmoid activation functions.

We have observed that reducing the angle bias causes an unfavorable side effect in the training of MLPs with rectified linear activation functions, in which the gradient exploding problem occurs. We need further studies to make LCW applicable to rectifier networks. Future work also includes investigating the applicability of LCW for other neural network structures, such as convolutional or recurrent structures. The use of LCW in recurrent networks is of particular interest, for which batch normalization is not straightforwardly applicable.

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

## A  SAMPLING UNIFORMLY FROM $\mathcal{W}_{\mathrm{LC1}}$

We can generate a sample $w$ that is uniformly distributed in $\mathcal{W}_{\mathrm{LC1}}$ as follows:

1. $w \sim N(0, I_m)$: Generate a sample $w$ from $m$-dimensional normal distribution $N(0, I_m)$.

2. $w \leftarrow w - E(w)$: Subtract the mean of elements of $w$, which corresponds to the projection of $w$ onto $\mathcal{W}_{\mathrm{LC}}$.

3. $w \leftarrow w/\|w\|$: Normalize $w$ to a unit vector.

4. $\xi \sim U(0, 1)$: Generate a sample $\xi$ from a uniform distribution in $(0, 1)$.

5. $w \leftarrow \xi^{\frac{1}{m-2}} w$: Shrink $w$ in proportion to the hypersurface area of the $m-1$ dimensional hypersphere of radius $\xi$.

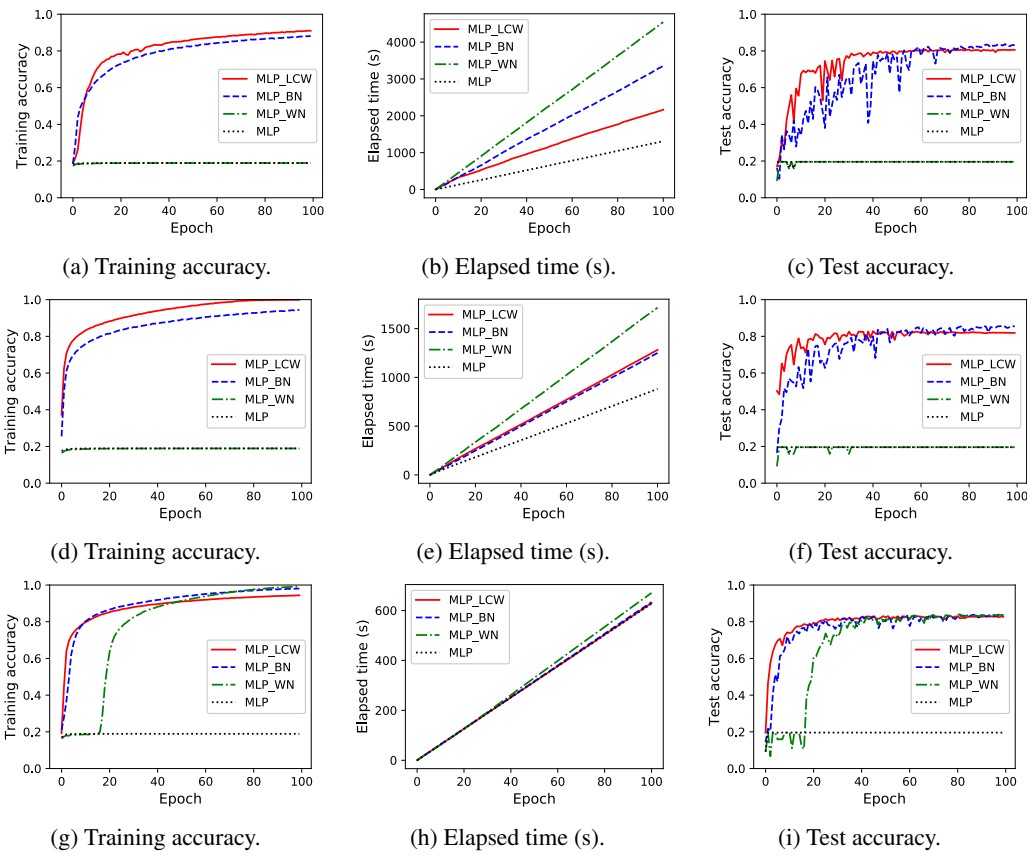

| (a) Training accuracy. | (b) Elapsed time (s). | (c) Test accuracy. |
|---|---|---|
| (d) Training accuracy. | (e) Elapsed time (s). | (f) Test accuracy. |
| (g) Training accuracy. | (h) Elapsed time (s). | (i) Test accuracy. |

Figure 14: Training accuracy, elapsed time, and test accuracy for SVHN dataset: (a-c) results of MLP$(100, 128)$, (d-f) results of MLP$(50, 256)$, (g-i) results of MLP$(5, 512)$. Sigmoid activation functions are used in each model.

## B    ADDITIONAL GRAPHS

The convergence and computation time for training MLPs with SVHN and CIFAR-100 datasets are shown in Figure 14 and Figure 15, respectively. In each figure, (a)-(c) shows results for MLP$(100, 128)$, (d)-(f) for MLP$(50, 256)$, and (g)-(i) for MLP$(5, 512)$. We see that both LCW and BN enable training deep models, such as MLP$(100, 128)$ and MLP$(50, 256)$. LCW achieves faster convergence with respect to the training accuracy in most cases.

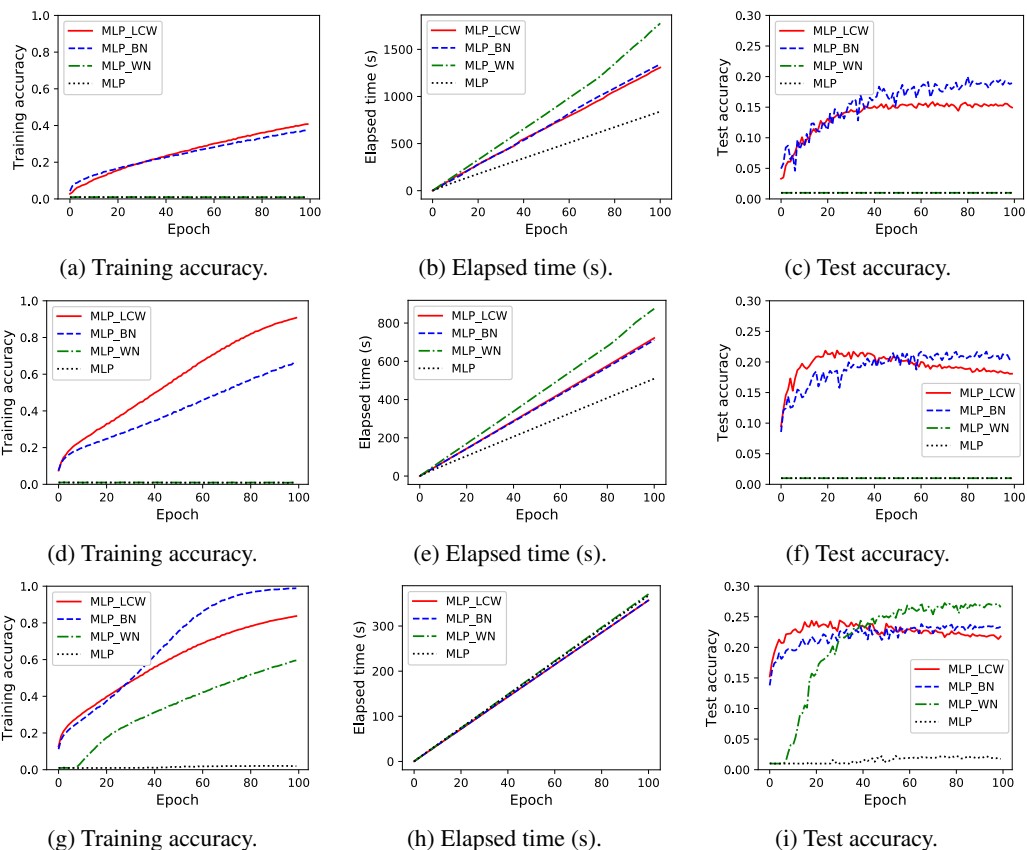

Figure 15: Training accuracy, elapsed time, and test accuracy for CIFAR-100 dataset: (a-c) results of MLP$(100, 128)$, (d-f) results of MLP$(50, 256)$, (g-i) results of MLP$(5, 512)$. Sigmoid activation functions are used in each model.

