# OpenReview forum: "Linearly Constrained Weights: Resolving the Vanishing Gradient Problem by Reducing Angle Bias"
_ICLR.cc/2018/Conference — Reject_

### Official Review · AnonReviewer3 · 2017-11-21
**Studies an interesting phenomenon but with preliminary results**

**Rating:** 5
**Confidence:** 4

**Review:**

This paper studies the impact of angle bias on learning deep neural networks, where angle bias is defined to be the expected value of the inner product of a random vectors (e.g., an activation vector) and a given vector (e.g., a weight vector).  The angle bias is non-zero as long as the random vector is non-zero in expectation and the given vector is non-zero.  This suggests that the some of the units in a deep neural network have large values (either positive or negative) regardless of the input, which in turn suggests vanishing gradient.  The proposed solution to angle bias is to place a linear constraint such that the sum of the weight becomes zero.  Although this does not rule out angle bias in general, it does so for the very special case where the expected value of the random vector is a vector consisting of a common value.  Nevertheless, numerical experiments suggest that the proposed approach can effectively reduce angle bias and improves the accuracy for training data in the CIFAR-10 task.  Test accuracy is not improved, however.

Overall, this paper introduces an interesting phenomenon that is worth studying to gain insights into how to train deep neural networks, but the results are rather preliminary both on theory and experiments.

On the theoretical side, the linearly constrained weights are only shown to work for a very special case.  There can be many other approaches to mitigate the impact of angle bias.  For example, how about scaling each variable in a way that the mean becomes zero, instead of scaling it into [-1,+1] as is done in the experiments?  When the mean of input is zero, there is no angle bias in the first layer.  Also, what about if we include the bias term so that b + w a is the preactivation value?

On the experimental side, it has been shown that linearly constrained weights can mitigate the impact of angle bias on vanishing gradient and can reduce the training error, but the test error is unfortunately increased for the particular task with the particular dataset in the experiments.  It would be desirable to identify specific tasks and datasets for which the proposed approach outperforms baselines.  It is intuitively expected that the proposed approach has some merit in some domains, but it is unclear exactly when and where it is.

Minor comments:

In Section 2.2, is Layer 1 the input layer or the next?

---

> ### Author Response · Authors · 2017-12-26
> **Responses to Reviewer3**
>
> We thank the reviewer for the insightful comments on our paper.
>
> --
>
> Comment 1: How about scaling each variable in a way that the mean becomes zero, instead of
> scaling it into [-1,+1] as is done in the experiments?  When the mean of input is zero,
> there is no angle bias in the first layer.
>
> Response 1: We did experiments with CIFAR-10, in which each variable was scaled to have
> zero mean. As the reviewer pointed out, we have no angle bias in the first layer (the layer
> after the input layer) in this case.
> However, the training of MLPs then got harder and the test accuracy was very row, even if
> we applied either LCW or batch-normalization. We think this is because normalizing each pixel
> of images in CIFAR-10 ruined the relationship between pixels.
>
> --
>
> Comment 2: What about if we include the bias term so that b + w a is the preactivation value?
>
> Response 2: We have already included the bias term in our original experiment, although
> it was omitted in Equation 2 for simplicity. We have modified Equation 2 to include
> the bias term for clarity in the revised manuscript.
>
> --
>
> Comment 3: It would be desirable to identify specific tasks and datasets for which
> the proposed approach outperforms baselines. It is intuitively expected that the proposed
> approach has some merit in some domains, but it is unclear exactly when and where it is.
>
> Response 3: We did additional experiments with the SVHN dataset and the CIFAR-100 dataset,
> which are reported in the appendix B of the revised manuscript. The peak value of the test
> accuracy of the proposed method was comparable to that of batch-normalization when the MLP
> has 5 layers or 50 layers, as shown in Figure 12 (f) and (i), Figure 14 (f) and (i), and
> Figure 15 (f) and (i).
> An interesting point is that the peak of the test accuracy is around 20 epochs in the
> proposed method. However, we have no clear explanation for this finding. We have added
> a description on this point in the third paragraph of Section 5.1 in the revised manuscript.
>
> --
>
> Comment 4: In Section 2.2, is Layer 1 the input layer or the next?
>
> Response 4: Layer 1 is the layer next to the input layer. We have added an explanation of
> these points to the first paragraph of Section 2.2.1 in the revised version.

---

> > ### Comment · AnonReviewer3 · 2018-01-10
> > **still not sufficiently strong**
> >
> > I appreciate the responses, additional experiments, and revision.  I see some advantages of the proposed LCW in test during early epochs, but those advantages are not quite strong.  Perhaps, more advantages might be seen in completely different domains or with better regularization methods.

---

### Official Review · AnonReviewer2 · 2017-11-27
**more disadvantages vs few advatages as of now.**

**Rating:** 5
**Confidence:** 4

**Review:**

The authors introduce the concept of angle bias (angle between a weight vector w and input vector x)  by which the resultant pre-activation (wx) is biased if ||x|| is non-zero or ||w|| is non-zero (theorm 2 from the article). The angle bias results in almost constant activation independent of input sample resulting in no weight updates for error reduction.   Authors chose to add an additional optimization constraint LCW (|w|=0) to achieve zero-mean pre-activation while, as mentioned in the article, other methods like batch normalization BN tend to push for |x|=0 and unit std to do the same.

Clearly, because of lack of scaling factor incase of LCW, like that in BN, it doesnot perform well when used with ReLU. When using with sigmoid the activation being bouded (0,1) seems to compensate for the lack of scaling in input. While BN explicitly makes the activation zero-mean LCW seems to achieve it through constraint on the weight features. Though it is shown to be computationally less expensive LCW seems to work in only specific cases unlike BN.

---

> ### Author Response · Authors · 2017-12-26
> **Responses to Reviewer2**
>
> We thank the reviewer for taking the time to evaluate our paper.
>
> --
>
> Comment 1: The authors introduce the concept of angle bias (angle between a weight vector w
> and input vector x)  by which the resultant pre-activation (wx) is biased if ||x|| is non-zero
> or ||w|| is non-zero (theorem 2 from the article). The angle bias results in almost constant
> activation independent of input sample resulting in no weight updates for error reduction.
> Authors chose to add an additional optimization constraint LCW (|w|=0) to achieve zero-mean
> pre-activation while, as mentioned in the article.
>
> Response 1: We did not intend to indicate that the proposed method (LCW) adds additional
> constraint ||w||=0 on weight vectors, and we have added an explanation to clearly state that
> it is assumed that ||w|| > 0 in our paper to the first paragraph of Section 2.1 in the
> revised manuscript.
> The proposed method adds constraints 'w_1 + .. + w_m = 0' on weight vectors w, where
> w = (w_1, ..., w_m)^\top in R^m, to force w perpendicular to 1_m = (1, ..., 1) in R^m,
> which is assumed to be the mean vector of the activation vector in the previous layer.
>
> --
>
> Comment 2: Clearly, because of lack of scaling factor in case of LCW, like that in BN,
> it does not perform well when used with ReLU. When using with sigmoid the activation being
> bounded (0,1) seems to compensate for the lack of scaling in input.
>
> Response 2: As the reviewer pointed out, the lack of scaling factor in LCW is a cause
> for not performing well with ReLU. We tried ReLU6 (= min(max(x, 0), 6)) instead
> of ReLU with LCW, but it was still hard to train a deep MLP, in which the exploding gradient
> still occurred. We are now developing methods to make LCW applicable to ReLU nets.
>
> --
>
> Comment 3: While BN explicitly makes the activation zero-mean LCW seems to achieve it through
> constraint on the weight features. Though it is shown to be computationally less expensive
> LCW seems to work in only specific cases unlike BN.
>
> Response 3: We agree that LCW has limitation compared to BN as of now. However, it is also
> very important to understand why batch normalization works so well in many situations.
> We believe that reducing angle bias is a crucial role of batch normalization, and such
> interpretation helps us to determine in which part of the network we should apply methods
> like batch normalization.

---

### Official Review · AnonReviewer1 · 2017-12-05
**This paper raises the concept of "angle bias" and introduces the so-called LCW method to reduce angle bias. The paper implies underlying connections between angle bias and the gradient vanishing problem and suggests that LCW is a cure for both issues. Although vanishing gradients in deep networks is an interesting topic for the community, the manuscript does not bring any novel theoretical understanding and there is also not enough empirical evidence to backup the claims made in the paper.**

**Rating:** 4
**Confidence:** 4

**Review:**

Pros:
The paper is easy to read. Logic flows naturally within the paper.

Cons:

1. Experimental results are neither enough nor convincing.

Only one set of data is used throughout the paper: the Cifar10 dataset, and the architecture used is only a 100 layered MLP. Even though LCW performs better than others in this circumstance, it does not prove its effectiveness in general or its elimination of the gradient vanishing problem. For the 100 layer MLP, it's very hard to train a simple MLP and the training/testing accuracy is very low for all the methods. More experiments with different number of layers and different architecture like ResNet should be tried to show better results.

In Figure (7), LCW seems to avoid gradient vanishing but introduces gradient exploding problem.

The proposed concept is only analyzed in MLP with Sigmoid activation function. In the experimental parts, the authors claim they use both ReLU and Sigmoid function, but no comparisons are reflected in the figures.

2. The whole standpoint of the paper is quite vague and not very convincing.
In section 2, the authors introduce angle bias and suggest its effect in MLPs that with random weights, showing that different samples may result in similar output in the second and deeper layers. However, the connection between angle bias and the issue of gradient vanishing lacks a clear analytical connection. The whole analysis of the connection is built solely on this one sentence "At the same time, the output does not change if we adjust the weight vectors in Layer 1", which is nowhere verified.

Further, the phenomenon is only tested on random initialization. When the network is trained for several iterations and becomes more settled, it is not clear how "angle affect" affects gradient vanishing problem.


Minors:
1. Theorem 1,2,3 are direct conclusions from the definitions and are mis-stated as Theorems.

2. 'patters' -> 'patterns'

3. In section 2.3, reasons 1 and 2 state the similar thing that output of MLP has relatively small change with different input data when angle bias occurs. Only reason 1 mentions the gradient vanishing problem, even though the title of this section is "Relation to Vanishing Gradient Problem".

---

> ### Author Response · Authors · 2017-12-26
> **Responses to Reviewer1**
>
> We thank the reviewer for the insightful comments on our paper.
>
> --
>
> Comment 1: Only one set of data is used throughout the paper: the Cifar10 dataset, and
> the architecture used is only a 100 layered MLP.
>
> Response 1: We did additional experiments with the SVHN dataset and the CIFAR-100 dataset
> for each of which we trained 5 layered, 50 layered, and 100 layered MLPs.
> Results are shown in Figure 12, Figure 14, and Figure 15 in the revised manuscript.
>
> --
>
> Comment 2: For the 100 layer MLP, it's very hard to train a simple MLP and the
> training/testing accuracy is very low for all the methods.
>
> Response 2: We do not agree to the comment. The training accuracy for CIFAR-10 or SVHN dataset
> is high for the 100 layer MLP, if we apply LCW (proposed method) or batch normalization,
> as shown Figure 12 (a) and Figure 14 (a) in the revised manuscript.
>
> --
>
> Comment 3: More experiments with different number of layers and different architecture
> like ResNet should be tried to show better results.
>
> Response 3: As mentioned in Response 1, we did experiments with several sizes of MLPs.
> We also tried ResNet, but it was unable to train ResNet with LCW. This is mainly because
> ReLU is used in ResNet, and the gradient explosion explained in Section 5.2 occurs.
> We are now developing methods that make LCW applicable to ReLU nets, including ResNet.
>
> --
>
> Comment 4: In Figure (7), LCW seems to avoid gradient vanishing but introduces gradient exploding problem.
>
> Response 4: We agree to the comment. We have added an explanation on these points to the
> second paragraph of Section 6 in the revised manuscript.
>
>
> --
>
> Comment 5: The proposed concept is only analyzed in MLP with Sigmoid activation function.
> In the experimental parts, the authors claim they use both ReLU and Sigmoid function,
> but no comparisons are reflected in the figures.
>
> Response 5: We omitted results with ReLU in the figures, because MLPs with ReLU were not
> trainable at all when LCW is applied, as mentioned in Section 5.2.
>
> --
>
> Comment 6: In section 2, the authors introduce angle bias and suggest its effect in MLPs that
> with random weights, showing that different samples may result in similar output in the second
> and deeper layers. However, the connection between angle bias and the issue of gradient
> vanishing lacks a clear analytical connection. The whole analysis of the connection is built
> solely on this one sentence "At the same time, the output does not change if we adjust the
> weight vectors in Layer 1", which is nowhere verified.
>
> Response 6: We have enriched the explanation in Section 2.1 in the revised manuscript,
> denoting that the shrinking of the distribution of the angle between the weight vector and the
> activation vector is a reason for why the activation becomes almost constant in deep layers.
> Moreover, we have added analytical results in Section 2.3 that examine the relationship
> between the constant activation in deeper layers and the vanishing gradient of weights.
>
> --
>
> Comment 7: The phenomenon is only tested on random initialization. When the network is trained
> for several iterations and becomes more settled, it is not clear how "angle affect" affects
> gradient vanishing problem.
>
> Response 7: We have added Figures 8 and 9, which show the activation and the distribution of
> angles in a MLP with sigmoid activation, respectively, after 10 epochs training.
> We have also added discussions on these figures to the third paragraph of Section 3.1.1 in
> the revised manuscript.
>
> --
>
> Comment 8: Theorem 1,2,3 are direct conclusions from the definitions and are mis-stated as Theorems.
>
> Response 8: We have modified the manuscript to refer to these statements as propositions instead of theorems.
>
> --
>
> Comment 9: 'patters' -> 'patterns'
>
> Response 9: In accordance with the comment, we have modified the expression.
>
> --
>
> Comment 10: In section 2.3, reasons 1 and 2 state the similar thing that output of MLP has relatively
> small change with different input data when angle bias occurs. Only reason 1 mentions the gradient
> vanishing problem, even though the title of this section is "Relation to Vanishing Gradient Problem".
>
> Response 10: In accordance with the comment, we have deleted the second reason from the manuscript.
> Also, we have enriched the explanation related to reason 1, as mentioned in Response 6.

---

> > ### Comment · AnonReviewer1 · 2018-01-12
> > **Reply to the rebuttal**
> >
> > I appreciate the effort taken by the authors to add more experimental results and enriching the intuitions of LCW. However, as the experiments have further shown, LCW only shows consistently better training accuracy than BN on a simple dataset as CIFAR10, but not as good on the testing data (therefore poor generalization) or on a more sophisticated data as CIFAR100.  Meanwhile, as the authors have pointed out, it’s not clear how to add LCW with ResNet, and to RNN where the vanishing gradient is more significant, which again limits its advantages. Therefore, I feel it is appropriate to keep my rating.

---

### Author Response · Authors · 2017-12-27
**Additional modifications**

In addition to the modifications discussed in the responses to the reviewer comments, we have revised
our paper in the following way:

- Vectors and matrices are written in bold font.

- Figures 4 and 5 are added to show the effect of the angle bias in a 50 layer MLP with ReLU activations.
  Section 2.2.2 is added to discuss these figures.

- Figures 10 and 11 are added to show the effect of LCW (proposed method) in the MLP with ReLU activations.
  Section 3.1.2 is added to discuss these figures.

- In the second paragraph of Section 3.3, we have added an explanation that bias terms are initialized to
  zero in the proposed method.

---

### Decision · Program_Chairs · 2018-01-29
**ICLR 2018 Conference Acceptance Decision**

**Decision:**

Reject

**Comment:**

The paper identifies an interesting problem in sigmoid deep nets, addressed diffferently by batchnorm, and proposes a different simple fix. It shows empirically that constraining neuron's weights to sum to zero improves training of a 100 layers sigmoid MLP.
The work is currenlty limited in its theoretical contribution, and regarding the showcased practical interest of the method compared to batchnorm (it's not appplicable to RELUs and shows positive effect on optimization but not generalization).